# PPARα Agonist, MHY3200, Alleviates Renal Inflammation during Aging via Regulating ROS/Akt/FoxO1 Signaling

**DOI:** 10.3390/molecules26113197

**Published:** 2021-05-26

**Authors:** Min Jo Kim, Dae Hyun Kim, EunJin Bang, Sang Gyun Noh, Pusoon Chun, Takako Yokozawa, Hyung Ryong Moon, Hae Young Chung

**Affiliations:** 1Department of Pharmacy, College of Pharmacy, Pusan National University, Geumjeong-gu, Busan 46241, Korea; kiki10304@gmail.com (M.J.K.); bioimmune@hanmail.net (D.H.K.); eunjin2285@gmail.com (E.B.); 2Interdisciplinary Research Programme of Bioinformatics and Longevity Science, Department of Pharmacy, College of Pharmacy, Pusan National University, Geumjeong-gu, Busan 46241, Korea; rskrsk92@naver.com; 3College of Pharmacy, Inje University, Gimhae 50834, Gyeongsangnam-do, Korea; pusoon@inje.ac.kr; 4Graduate School Science and Engineering for Research, University of Toyama, Toyama 930-8555, Japan; yokozawa@inm.u-toyama.ac.jp

**Keywords:** aging, MHY3200, ROS, Akt, NF-κB, inflammation

## Abstract

PPARα is a ligand-dependent transcription factor and its activation is known to play an important role in cell defense through anti-inflammatory and antioxidant effects. MHY3200 (2-[4-(5-chlorobenzo[*d*]thiazol-2-yl)phenoxy]-2,2-difluoroacetic acid), a novel benzothiazole-derived peroxisome proliferator-activated receptor α (PPARα) agonist, is a synthesized PPARα activator. This study examined the beneficial effects of MHY3200 on age-associated alterations in reactive oxygen species (ROS)/Akt/forkhead box (FoxO) 1 signaling in rat kidneys. Young (7-month-old) and old (22-month-old) rats were treated with MHY3200 (1 mg/kg body weight/day or 3 mg/kg body weight/day) for two weeks. MHY3200 treatment led to a notable decrease in triglyceride and insulin levels in serum from old rats. The elevated kidney ROS level, serum insulin level, and Akt phosphorylation in old rats were reduced following MHY3200 treatment; moreover, FoxO1 phosphorylation increased. MHY3200 treatment led to the increased level of FoxO1 and its target gene, MnSOD. MHY3200 suppressed cyclooxygenase-2 expression by activating PPARα and inhibiting the activation of nuclear factor-κB (NF-κB) in the kidneys of old rats. Our results suggest that MHY3200 ameliorates age-associated renal inflammation by regulating NF-κB and FoxO1 via ROS/Akt signaling.

## 1. Introduction

Aging is characterized by physiological changes such as immunosenescence, decreased hormonal secretion and body mass, and lipid accumulation. The incidence of various diseases, such as inflammatory diseases, obesity, dyslipidemia, type 2 diabetes, and atherosclerosis, increases with age [1]. Significantly, most characteristics of the aging are associated with the activities of peroxisome proliferator-activated receptors (PPARs).

PPARs are important transcription factors that participate in a wide range of biological processes, including glucose homeostasis, prostaglandin cascade, proliferation, insulin signaling, lipid and glucose metabolism, tissue remodeling, and cell differentiation. Detailed studies on PPARs have demonstrated their critical roles in both physiological as well as diseased conditions of various tissues [2,3,4]. The PPAR subfamily comprises three isotypes: PPARα, PPARβ/δ, and PPARγ which exhibit tissue-specific expression due to differential expression in different tissues and organs. PPARα is overexpressed in the kidneys and liver, leading to enhanced lipid oxidation and expression of genes encoding antioxidant enzymes such as manganese superoxide dismutase (MnSOD) and catalase [5]. Therefore, PPARα activation plays a critical role in cellular defense through anti-inflammatory and antioxidant effects [6].

In our previous study, 2-[4-(5-chlorobenzo[*d*]thiazol-2-yl)phenoxy]-2,2-difluoroacetic acid (MHY3200), a novel benzothiazole-derived PPARα agonist, was assessed in AC2F cells using PPRE-luciferase assay, transcriptional activity, and docking simulation. Furthermore, in metabolic organ liver, MHY3200 ameliorated high-fat diet-induced inflammation and lipid accumulation in the liver and reduced insulin resistance via PPARα activation [7].

However, a non-metabolic organ such as kidney ages faster than other body organs. The expression of aging biomarkers differs based on oxidative stress and inflammation [8,9]. Age-associated oxidative damage increases in tandem with an increase in reactive oxygen species (ROS) [10,11,12]. Thus, regulation of oxidative damage plays a critical role in senescence control. In this study, we investigated whether MHY3200 alleviates inflammatory responses by modulating ROS/Akt/forkhead box O1 (FoxO1) signaling and PPARα activity in the kidneys of old rats.

## 2. Results

### 2.1. MHY3200 Reverses the Serum Biochemical Profile of Aged Rats

Dyslipidemia is associated with aging-related diseases and is characterized by increased serum TG and total cholesterol levels [13]. The present study demonstrated that serum TG levels were higher by 2.12-fold in old rats compared with young rats. These levels were significantly reduced by 5.1-fold and by 4.4-fold following treatment with both 1 mg and 3 mg of MHY3200, respectively (Figure 1A). In addition, total serum cholesterol levels were higher by 1.82-fold in the serum of old rats compared with their young counterparts, but treatment with 1 mg and 3 mg MHY3200 decreased these levels by 1.85-fold and 2.25-fold, respectively (Figure 1B). No significant differences were observed in serum glucose levels in rats of the different groups (Figure 1C). Insulin levels were higher by 2.02-fold in old rats than in young rats; however, the levels decreased significantly by 3.4-fold and by 3.59-fold following treatment with 1 mg and 3 mg MHY3200, respectively (Figure 1D). These results indicate that MHY3200 exerts beneficial effects on lipid parameters during aging.

### 2.2. Changes in NOX4/Akt/FoxO1 Signaling in Aged Kidneys

Oxidative stress-associated expression of NADPH oxidase subunit 4 (NOX4) protein was significantly enhanced in the kidneys of old rats, but MHY3200 administration successfully reduced this enhanced protein expression (Figure 2A). Furthermore, similar to the expression of NOX4 protein, the levels of ROS were notably higher in the kidneys of old rats than in those of young rats, although these elevated levels were significantly suppressed by administration of 1 mg/kg or 3 mg/kg MHY3200 (Figure 2B). In addition, the ROS-mediated phosphorylation of the serine residue in Akt was enhanced in old rats compared with young rats, but MHY3200 administration normalized this phosphorylation (Figure 2A). Western blotting was performed using cytosolic and nuclear fractions of kidney proteins to examine the effects of MHY3200 on FoxO1 level and FoxO1-targeted gene expression in the kidneys of old rats. Phosphorylation of FoxO1 serine256 residue and the expression of nuclear FoxO1 were upregulated in aged rats (Figure 2C). Moreover, the expression of MnSOD decreased in old rats compared with young rats, leading to increased ROS levels and inflammation in aged rats. However, the administration of MHY3200 reversed this effect (Figure 2D). These data indicate that MHY3200 attenuated NOX4-related ROS production and Akt and FoxO1 phosphorylation in the kidneys of old rats, leading to a decrease in oxidative stress and inflammation.

### 2.3. Effects of MHY3200 on Nuclear Factor-kappa B (NF-κB) and Its Inflammatory Signaling during Aging

Since MHY3200 showed a beneficial effect on oxidative stress, we further investigated the expression of oxidative stress-associated proteins such as NF-κB and its target protein, COX-2. Old rats showed increased nuclear levels of phosphorylated NF-κB protein and its target COX-2, compared with young rats, but the administration of MHY3200 successfully reduced NF-κB activity and COX-2 protein expression (Figure 3A,B). These findings suggest that MHY3200 decreases age-associated renal inflammatory responses via inhibiting NF-κB activity.

### 2.4. MHY3200 Binding Affinity to PPARα and Its Effects on PPARα Activity during Aging

We conducted an in silico docking simulation to investigate whether MHY3200 can bind directly to PPARα with high affinity. The molecular docking models of MHY3200 (blue) and a well-known PPARα agonist, WY14643 (red) in the binding site of PPARα, are shown in Figure 4A, together with their binding energy. The MHY3200-PPARα complex had −8.89 kcal/mol binding energy, which was higher than that of WY14643 (Figure 4A). The molecular interactions between PPARα and MHY3200 were visualized using the Autodock 4.2.6 program (Scripps Research, San Diego, California, US) to investigate whether MHY3200 could bind to PPARα. The possible residues involved in van der Waals interactions between MHY3200 and PPARα included PHE273, CYS276, TYR314, LEU344, LEU347, PHE351, ILE354, MET355, HIS440, LEU456, and LEU460. One additional critical interactive residue, TYR464, formed a hydrogen bond (shown in red square) (Figure 4B). On the other hand, WY14643, which was used as a control in this study, has multiple interactive residues with PPARα, including ILE272, PHE273, CYS276, GLN277, THR279, SER280, PHE318, LEU321, LEU344, PHE351, ILE354, and MET355. However, WY14643 did not form a hydrogen bond with PPARα (Figure 4C), indicating that MHY3200 has a higher binding affinity for PPARα than WY14643. Furthermore, nuclear levels of PPARα protein were increased in the kidneys of old rats treated with MHY3200 (Figure 4D). These results suggest that MHY3200 is a potential PPARα agonist that can regulate inflammation and attenuating age-related inflammatory responses via modulating PPARα activation in the kidneys of old rats.

## 3. Discussion

The free radical theory of aging describes the principal mechanisms of aging and the pathogenesis of diverse diseases that follow aging, including arthritis, diabetes, osteoporosis, dementia, and atherosclerosis [14]. To investigate the potential association between the free radical theory and age-associated diseases, several studies have focused on elevated ROS levels, which disrupt the redox balance, leading to sustained chronic inflammatory responses [15].

During aging, inflammatory responses are hyperactivated, resulting in tissue damage and leading to detrimental disease conditions [16]. Reactive oxygen and nitrogen species associated with defective antioxidant defense capability leads to redox imbalance, which induces the activation of redox-dependent transcription factors that regulate the transcription of diverse proinflammatory mediators [17]. The increased ROS levels observed during aging induce Akt activation. The inability of aged kidneys to suppress oxidative stress predisposes them to ROS-mediated inflammation [18]. In addition, FoxO isoforms induce the expression of downstream target genes associated with the regulation of oxidative stress, the cell cycle, and metabolism. [19]. FoxO activity is modulated by phosphoinositide 3-kinase (PI3K)/protein kinase B (Akt) signaling [20] in aged rat kidneys. Activated Akt inhibits FoxO activity, leading to a decrease in the expression of FoxOs-mediated antioxidant enzymes such as MnSOD and catalase, which upregulate the expression of genes associated with NF-κB-dependent proinflammatory mediators [21]. Moreover, Akt-regulated FoxOs can suppress intracellular ROS levels by enhancing the expression of MnSOD [22]. Akt-induced phosphorylation of FoxO1 causes shuttling of FoxO1 from the nucleus to the cytosol, leading to increased cellular ROS [23,24]. In the current study, aged rats were sensitive to oxidative damage, while MHY3200 treatment inhibited Akt phosphorylation and increased FoxO1 activation, leading to the upregulated expression of the gene encoding the antioxidant MnSOD (Figure 2).

NF-κB regulates inflammatory signaling caused by oxidative stress during aging [25]. NF-κB is a ubiquitous transcription factor stimulated by diverse stimuli, including inflammation, infection, and oxidative stress. NF-κB activation during renal aging induces the expression of pro-inflammatory enzymes such as COX-2, inducible nitric oxide synthase (iNOS), as well as several cytokines [26]. The upregulation of inflammatory genes leads to chronic pro-inflammatory conditions [27]. The present study demonstrated that NF-κB activation and COX-2 protein expression were notably upregulated in the kidneys of aged rats. However, administration of MHY3200 significantly inhibited the expression of these proteins (Figure 3). These findings indicate that NF-κB and its target gene, COX-2, may play important roles in kidney diseases, including age-related kidney diseases.

We previously found that the synthesized 2-[4-(5-chlorobenzo[*d*]thiazol-2-yl) phenoxy]-2-methylpropionic acid (MHY908) acts as a PPARα/γ dual agonist [28]. To elucidate the mechanism of action of MHY908, Kim et al. [29] investigated whether MHY908 suppresses inflammation by regulating Akt/FoxO1 signaling in the kidneys of aged rats. They found that administering MHY908 to old rats inhibited IRS/Akt phosphorylation and activated FoxO1, which upregulated the expression of both catalase and MnSOD encoding genes in the kidneys of old rats. However, MHY908 treatment reduced NF-κB signaling in insulin-treated cells. These findings, which are in line with the effects observed in the present study, suggest that MHY908 decreases age-related kidney inflammation by regulating ROS/Akt/FoxO1 signaling and NF-κB activity. In contrast, PPARα activity notably increased in a dose-dependent manner by MHY3200 treatment [7]. In addition, the docking simulation-based prediction of the affinity of MHY3200 and PPARα binding was done by the AutoDock 4.2.6 program and the binding affinity was –8.89 kcal/mol. However, the binding affinity of MHY3200 based on molecular docking simulation studies is in accordance with PPARα level in MHY3200-treated rat kidneys (Figure 4). These data suggest that MHY3200 is a potential PPARα agonist that can modulate age-related kidney inflammation.

In conclusion, the administration of MHY3200 exerts diverse effects on several serum parameters that modulate the progression of aging. Overall, MHY3200 effectively ameliorated renal inflammation by regulating the ROS/Akt/FoxO1 pathway and exerted beneficial effects on renal disorders caused by aging (Figure 5). These findings indicate that MHY3200 improves renal inflammation by inhibiting ROS and Akt and enhancing FoxO1 activity and MnSOD expression in aging-related progressive renal injury.

## 4. Materials and Methods

### 4.1. Materials

MHY3200 was designed and synthesized by Prof. H. R. Moon at the College of Pharmacy, Pusan National University, Busan, Korea [7], and dissolved in dimethyl sulfoxide (DMSO). Other chemical reagents were obtained from Sigma–Aldrich (St. Louis, MO, USA). 2’,7’-Dichlorodihydrofluorescein diacetate (DCFH-DA) was purchased from Molecular Probes Inc. (Eugene, OR, USA). ECL Western Blotting Detection Reagents, Pierce bicinchoninic acid (BCA) protein assay kit, polyvinylidene difluoride (PVDF) membrane, and Dokdo-MARK™ protein size marker were purchased from GE Healthcare Biosciences (Buckinghamshire, UK), Thermo Fisher Scientific (Rockford, IL, USA), the Millipore Corporation (Bedford, MA, USA), and ElpisBiotech (Daejeon, Korea), respectively. Anti-NOX4, anti-p-Akt (Ser 473), manganese superoxide dismutase (MnSOD), p-p65 (Ser 276), p65, β-actin, transcription factor IIB (TFIIB), and cyclooxygenase-2 (COX-2) were purchased from Santa Cruz Biotechnology (Dallas, TX, USA). Antibodies against p-FoxO1 (Ser 256) were purchased from Cell Signaling Technology (Danvers, MA, USA) while anti-rabbit IgG-horseradish peroxidase-conjugated antibodies, anti-mouse IgG-horseradish peroxidase-conjugated antibodies, and anti-goat IgG-horseradish peroxidase-conjugated antibodies were purchased from Santa Cruz Biotechnology (Dallas, TX, USA). 

### 4.2. Animal Experiments

In this study, 7-month-old and 22-month-old Sprague Dawley male rats were used to examine the beneficial effects of MHY3200 treatment on the inflammatory response during aging. The rats were supplied by Samtako (Gueonggi-do, Korea) and stabilized at the animal care facility for seven days prior to conducting experiments. Animals were housed in an air-conditioned room under a 12 h light/dark cycle with access to water and standard rodent chow (Samtako). The animal experiments were reviewed and approved by the Institutional Animal Care Committee of Pusan National University (Approval Number PNU-2017-1534) and performed according to the stipulated guidelines. Water vehicle or MHY3200 (1 or 3 mg/kg/day) was orally administered to rats for two weeks. Food consumption of rats was monitored daily and their body weights were measured once every three days. Food intake and increase in body weight were similar among the groups.

### 4.3. Biochemical Analyses of Rat Serum

Blood samples were collected from animals in each group after they were sacrificed. Serum samples were collected via centrifugation at 2000 × *g* at 4 °C for 15 min. Serum concentrations of triglyceride (TG), total cholesterol, and glucose were analyzed using kits from Shinyang Chemical Co. (Seoul, Korea). Specific kits were used to measure insulin concentrations (Shibayagi, Japan), following the manufacturer’s protocol.

### 4.4. Western Blot Analyses

Western blot analysis was performed as previously described [30]. Briefly, protein samples extracted from the kidneys were boiled for 5 min in gel-loading buffer (0.125 M Tris-HCl (pH 6.8), 4% sodium dodecyl sulfate (SDS), 10% 2-mercaptoethanol, and 0.2% bromophenol blue) at a ratio of 1:1. An equal quantity of protein was subjected to SDS-polyacrylamide gel electrophoresis on 6–17% acrylamide gels. The gels were subsequently transferred to Immobilon-P transfer polyvinylidene fluoride membranes (Millipore Corp, Bedford, MA, USA). The membranes were immediately placed in blocking buffer composed of 5% non-fat milk in TBS-Tween buffer (TBS-T, 10 mM Tris, 0.1% Tween 20; pH 7.5) and 100 mM NaCl and incubated at room temperature for 1 h, after which the membranes were washed with TBS-T buffer for 30 min and incubated overnight at 4 °C with specific primary antibodies. Next day, the membranes were washed with TBS-T buffer, incubated with a secondary antibody for 1 h at room temperature, and then washed again with TBS-T buffer. Finally, the protein bands were detected using secondary antibody and ECL kit according to the manufacturer’s instructions.

### 4.5. Preparation of Cytosolic and Nuclear Fraction Homogenates from Kidney Tissue

We homogenized approximately 100 mg of frozen kidney tissue with 1 mL homogenate buffer solution (pH 7.4) containing 10 mM HEPES, 20 mM β-glycerophosphate, 20 mM NaF, 2 mM sodium orthovanadate, 0.5 mM phenylmethylsulfonyl fluoride (PMSF) (pH 7.4), 1× protease inhibitor cocktail solution, 1 mM EDTA, and 0.01 mM dithiothreitol (DTT) in a tissue homogenizer for 30 s, and incubated the homogenate on ice for 30 min. A total of 87.5 μL of 10% Nonidet P-40 (NP-40) solution was added to the homogenate, mixed for 15 s, and centrifuged at 12,000× *g* for 5 min at 4 °C. The supernatant was collected as the cytosolic fraction. The pellet was washed once with 200 μL homogenate buffer solution and 25 μL 10% NP-40, centrifuged and resuspended in 100 μL buffer containing 50 mM KCl, 300 mM NaCl, 0.1 mM EDTA, 10% (*v/v*) glycerol, 0.01 mM DTT, 20 mM β-glycerophosphate, 20 mM NaF, 2 mM sodium orthovanadate, 1 mM EDTA, 0.5 mM PMSF, and 1× protease inhibitor cocktail solution. This suspension was incubated on ice for 30 min and then centrifuged at 12,000× *g* for 10 min at 4 °C. The supernatant was collected as the nuclear fraction and stored at −80 °C. 

### 4.6. ROS Measurement

ROS production was measured using a fluorescent probe, DCFH-DA. Briefly, DCFH-DA (25 μM) was solubilized in 50 mM phosphate buffer. Alterations in fluorescence intensity were measured every 5 min for 30 min using a microplate reader (Berthold Technologies, GmbH & Co., Bad Wildbad, Germany), with excitation and emission wavelengths of 485 and 530 nm, respectively.

### 4.7. In Silico Protein-Ligand Docking Simulation

The crystal structure of human PPARα was obtained from the Protein Data Bank (PDB) archives (https://www.rcsb.org/, entry code PPARα: 1K7L) and used as a target for the docking simulation performed using the AutoDock 4.2.6 program (OpenEye Scientific Software, SantaFe, NM, USA). A predefined active site in human PPARα was used to define the docking pocket of PPARα and a docking simulation between PPARα and MHY3200 was performed. Their molecular interactions were visualized using AutoDock 4.2.6 program. To prepare compounds for the docking simulation, 2D structures were first converted into 3D structures. Subsequently, charges were calculated, and hydrogen atoms were added using the ChemOffice program (http://www.cambridgesoft.com). 

### 4.8. Statistical Analysis

All results are presented as mean ± standard error of the mean (SEM). All groups were compared using one-way analysis of variance (ANOVA) and tested for statistical significance using the Bonferroni test. Statistical significance was set at *p* < 0.05. The analysis was performed in GraphPad Prism 5 (GraphPad Software, La Jolla, CA, USA).

## Figures and Tables

**Figure 1 molecules-26-03197-f001:**
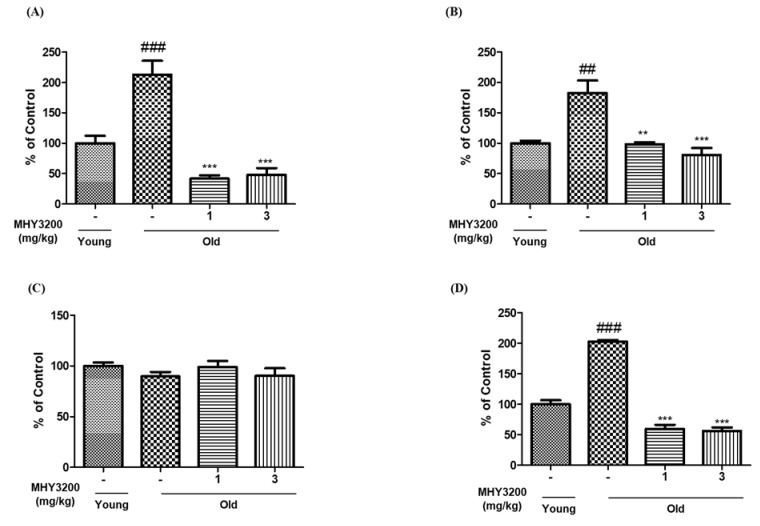
Biochemical features of aging. MHY3200 (1 mg/kg/day) or (3 mg/kg/day) was orally administered for 14 days in old rats. (**A**) triglyceride, (**B**) total cholesterol, (**C**) glucose, (**D**) insulin concentration in serum was quantified using the colorimetric kit. Young (6-month-old) rats, Young; Old (22-month-old) rats, Old. One-way analysis of variance (ANOVA) was used to determine the statistical result: ^##^
*p* < 0.01, ^###^
*p* < 0.001 vs. Young rats; ^**^
*p* < 0.01, ^***^
*p* < 0.001 vs. Old rats. Young (-), non-treated young rats; Old (-), non-treated old rats; 1, MHY3200 (1mg/kg)-treated old rats; 3, MHY3200 (3mg/kg)-treated old rats.

**Figure 2 molecules-26-03197-f002:**
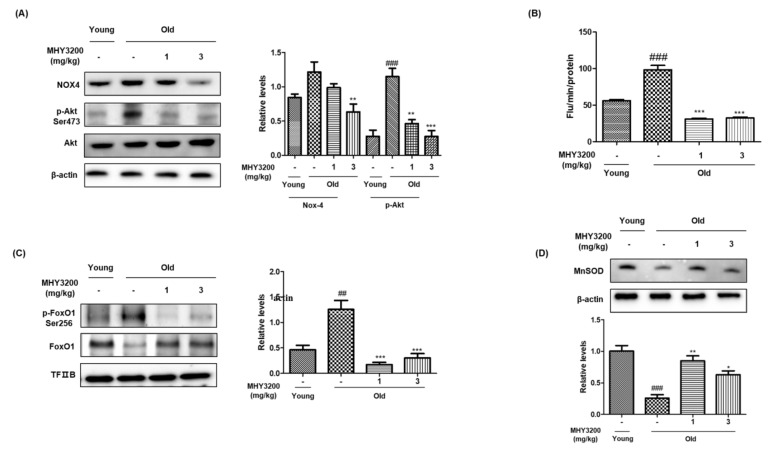
Effect of MHY3200 on NOX4/Akt signaling during the aging process. (**A**) Western blot analysis was performed to detect NOX4, phosphorylated Akt, and total-Akt in cytoplasmic extracts from kidneys. (**B**) ROS production was measured in the kidneys of young and aged rats using DCFH-DA assay. (**C**) Western blot analysis was performed to detect the phosphorylation of FoxO1 serine256 residue and total FoxO1 in the nuclear fraction. (**D**) Levels of MnSOD protein in the cytoplasmic extracts from rat kidneys. TFIIB and β-actin were used as nuclear and cytosolic internal controls and graphs were quantified with CS analyzer image analysis software. One representative blot is shown from three independent experiments in each group that yielded similar results (*n* = 6). One-way analysis of variance (ANOVA) was used to determine the statistical result: ^##^
*p* < 0.01, ^###^
*p* < 0.001 vs. Young rats; ^*^
*p* < 0.05, ^**^
*p* < 0.01, ^***^
*p* < 0.001 vs. Old rats. Young (-), non-treated young rats; Old (-), non-treated old rats; 1, MHY3200 (1mg/kg)-treated old rats; 3, MHY3200 (3mg/kg)-treated old rats.

**Figure 3 molecules-26-03197-f003:**
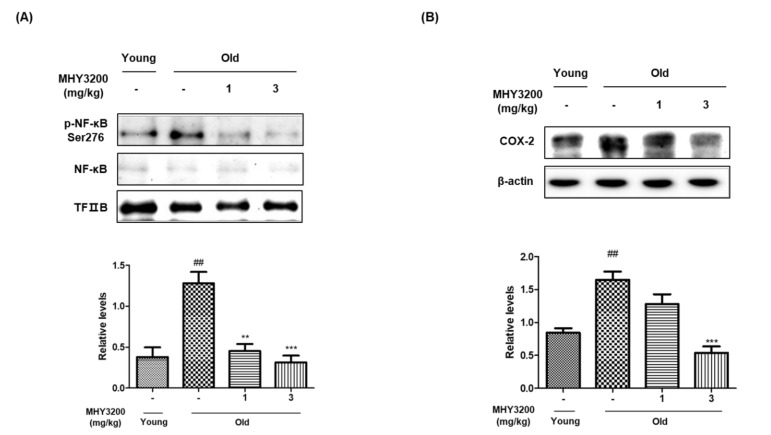
Effect of MHY3200 on NF-κB and its downstream inflammatory signaling during aging. (**A**) Nuclear fraction NF-κB and (**B**) cytosol fraction COX-2 protein levels were determined using Western blot analysis. TFIIB and β-actin were used as the nuclear and cytosolic internal controls and graphs were quantified with CS analyzer image analysis software. One representative blot is shown from three independent experiments in each group that yielded similar results (*n* = 6). One-way analysis of variance (ANOVA) was used to determine the statistical result: ^##^
*p* < 0.01 vs. Young rats; ^**^
*p* < 0.01, ^***^
*p* < 0.001 vs. Old rats. Young (-), non-treated young rats; Old (-), non-treated old rats; 1, MHY3200 (1mg/kg)-treated old rats; 3, MHY3200 (3mg/kg)-treated old rats.

**Figure 4 molecules-26-03197-f004:**
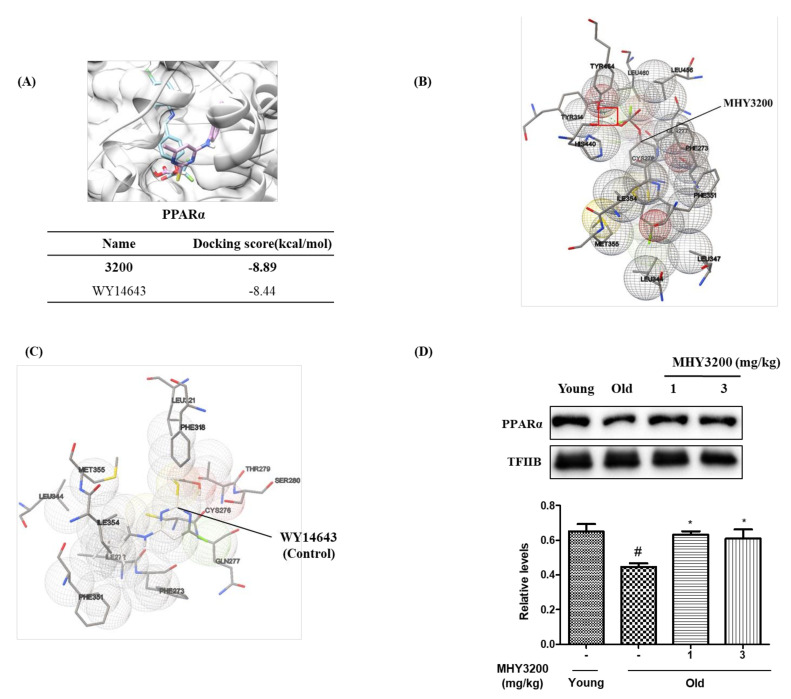
Binding affinity between MHY3200 and PPARα and its effects on PPARα activity during aging. Docking simulation was performed to identify the interaction between MHY3200 and PPARα and the ligand binding affinity was assessed. (**A**) MHY3200 has similar binding affinity to the known PPARα agonist, WY14643. (**B**) MHY3200 interacts with PPARα via multiple interactions involving several residues, including a single hydrogen bond (shown in the red square). (**C**) The control compound WY14643 also interacts with PPARα via multiple residues but does not form a hydrogen bond. (**D**) Levels of PPARα protein in the nucleus. TFIIB was used as an internal nuclear control and graphs were quantified with CS analyzer image analysis software. One representative blot is shown from three independent experiments in each group that yielded similar results (*n* = 6). One-way analysis of variance (ANOVA) was used to determine the statistical result: ^#^
*p* < 0.05 vs. Young rats; ^*^
*p* < 0.05 vs. Old rats. Young (-), non-treated young rats; Old (-), non-treated old rats; 1, MHY3200 (1mg/kg)-treated old rats; 3, MHY3200 (3mg/kg)-treated old rats.

**Figure 5 molecules-26-03197-f005:**
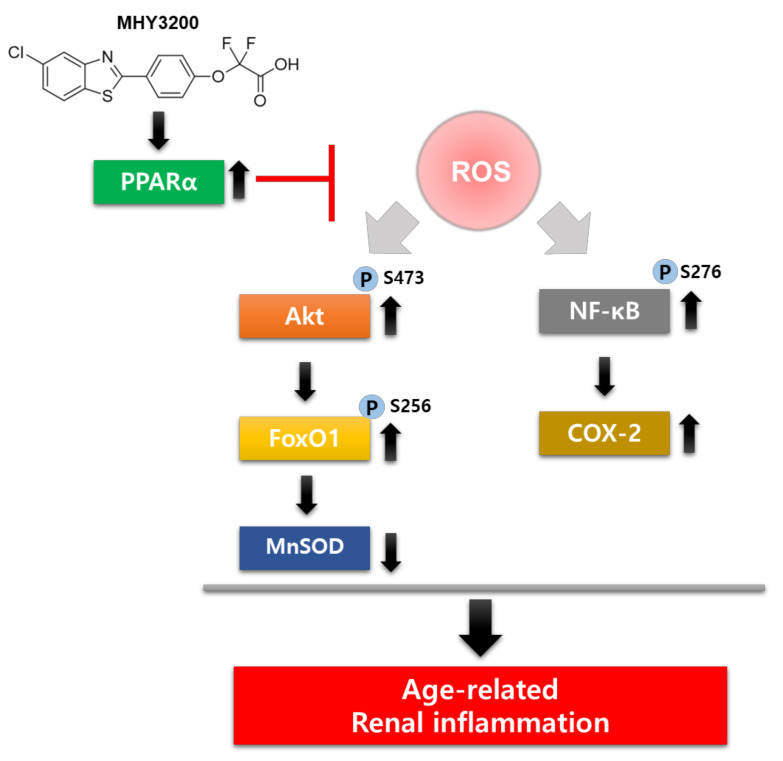
Possible role of MHY3200 in modulating ROS/Akt/FoxO1 signaling in kidney tissues of aged rats. MHY3200 ameliorated age-related inflammatory response by inhibiting the proinflammatory NF-κB activity regulated by ROS/Akt/FoxO1 signaling.

## Data Availability

Data is contained within the article.

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
