# Peer review of "PPARα Agonist, MHY3200, Alleviates Renal Inflammation during Aging via Regulating ROS/Akt/FoxO1 Signaling"

_molecules, 2021, doi:10.3390/molecules26113197_

Round 1
Reviewer 1 Report
Dear authors,
This article describes an interesting study on the effect of compound MHY3200 on aging.
Although the work is well presented I have several suggestions/doubts that I would like to be revised just to ease the reading of the manuscript.
-ABSTRACT:
- Taking into account that the abstract is the first part read by researchers, it must be concise, clear, and easy to follow. However, in this case, in my opinion, the abstract shows one sentence that can lead to confusion. So, in lines 22-23: “MHY3200 treatment led to a notable decrease in triglyceride and insulin levels, and a slight decrease in total cholesterol level increased in serum from old rats”.
It is not clear if cholesterol increases or decreases. Although It can be understood the meaning of the sentence, It is not clear if cholesterol increases or decreases, and a quick reading can lead to a wrong concept. Please, clarify or rewrite this sentence avoiding ambiguity.
- “MHY3200 is a newly synthesized…activator”. As the synthesis of this compound was published in 2018, I would not say “newly synthesized”.
-MANUSCRIPT:
- The first figure showed in the text is “Figure 2” while “Figure 1” appears in “Materials and Methods”. Please, renumber the figures as the first one must be “Figure 1”.
- Along the manuscript is not clear if the treatment with MHY3200 is carried out on either young and old rats or only in old rats. In the case that the treatment is on both types of rats, where are the results obtained for young rats?. Did I lose them?. Please, clarify!.
- Even when the figures are clear and the decreasing of the studied parameters in serum are easy to follow and significant, I think that it would be better to give the results as percentages.
- In figure 2, the format for figure 2 A-B and the format for figure 2 C-D are different, why?
Author Response
ABSTRACT:
Taking into account that the abstract is the first part read by researchers, it must be concise, clear, and easy to follow. However, in this case, in my opinion, the abstract shows one sentence that can lead to confusion. So, in lines 22-23: “MHY3200 treatment led to a notable decrease in triglyceride and insulin levels, and a slight decrease in total cholesterol level increased in serum from old rats”.
It is not clear if cholesterol increases or decreases. Although It can be understood the meaning of the sentence, It is not clear if cholesterol increases or decreases, and a quick reading can lead to a wrong concept. Please, clarify or rewrite this sentence avoiding ambiguity.
Response: Thanks for your comments. We delete it (line 31, page 2).
“MHY3200 is a newly synthesized…activator”. As the synthesis of this compound was published in 2018, I would not say “newly synthesized”.
Response: Thanks for your comments. We delete “newly” (line 26, page 2; line 163, page 9).
MANUSCRIPT:
The first figure showed in the text is “Figure 2” while “Figure 1” appears in “Materials and Methods”. Please, renumber the figures as the first one must be “Figure 1”.
Response: Thank you for your comment. Following the comment, it was corrected.
Along the manuscript is not clear if the treatment with MHY3200 is carried out on either young and old rats or only in old rats. In the case that the treatment is on both types of rats, where are the results obtained for young rats?. Did I lose them?. Please, clarify!.
Response: Thank you for your comment. The young rat was used as a control group for the old rat, and effects of MHY3200 treatment was examined in the old rats. There are no results on the young rats.
Even when the figures are clear and the decreasing of the studied parameters in serum are easy to follow and significant, I think that it would be better to give the results as percentages.
Response: Thank you for your comment. According to the comment, it was corrected.
In figure 2, the format for figure 2 A-B and the format for figure 2 C-D are different, why?
Response: Thank you for your comment. We modified it to the same format.
Reviewer 2 Report
The authors present an interesting paper on the role of PPARa agonist, 2-(4-(5-chlorobenzo[d]thiazol-2-yl)phenoxy)-2,2-difluoroacetic acid (MHY3200), in alleviating renal inflammation during aging via regulating ROS/AKT/FoxO1 signaling. Previously they had investigated the effects of MHY3200 on high-fat diet (HFD)-induced hepatic lipid accumulation and inflammation in rat, here they focused on rat old kidney.
The paper is experimentally well done but could be improved.
Major Revision
1) In Figure 3, 4 and 5 D histograms of western blot with error bars (SEM) and the statistical significance of the results should be added.
2) is PPARa protein reduced in old kidney rats?
3) the predicted binding affinity of MHY3200 and PPARα was yet described in the previous paper, please mention in this work.
4) Figure 6 should be graphically improved.
5) the text should be typographically checked.
Author Response
Major Revision
1) In Figure 3, 4 and 5 D histograms of western blot with error bars (SEM) and the statistical significance of the results should be added.
Response: Thank you for your comment. According to the comment, we add figure legends.
2) is PPARa protein reduced in old kidney rats?
Response: Thank you for your comment. We have confirmed the decline of nuclear PPARα protein in old rat kidneys. We checked the same result as the previous paper published in our lab.
Ref. 1) J Am Soc Nephrol. 2018 Apr;29(4):1223-1237. doi: 10.1681/ASN.2017070802.
Ref. 2) Exp Gerontol. 2017 Jun;92:87-95. doi: 10.1016/j.exger.2017.03.015.
3) the predicted binding affinity of MHY3200 and PPARα was yet described in the previous paper, please mention in this work.
Response: Thank you for your comment. Following the comment, it was corrected (line 115, page 6).
4) Figure 6 should be graphically improved.
Response: Thank you for your comment. Following the comment, it was corrected.
5) the text should be typographically checked.
Response: Thank you for your comment. Following the comment, it was corrected.
Round 2
Reviewer 2 Report
In this version the work has been improved and deserves to be publishedAuthor Response
Thank you for your comment. Following the comment, it was corrected.
I marked the modified part in red.